# Advancements in Forest Fire Prevention: A Comprehensive Survey

**DOI:** 10.3390/s23146635

**Published:** 2023-07-24

**Authors:** Francesco Carta, Chiara Zidda, Martina Putzu, Daniele Loru, Matteo Anedda, Daniele Giusto

**Affiliations:** CNIT UdR, Department of Electrical and Electronic Engineering, University of Cagliari, 09123 Cagliari, Italy; f.carta52@studenti.unica.it (F.C.); chiara.zidda@unica.it (C.Z.); m.putzu21@studenti.unica.it (M.P.); d.loru2@studenti.unica.it (D.L.)

**Keywords:** fire detection, terrestrial, aerial, satellite, artificial intelligence, deep learning, UAV, sensors

## Abstract

Nowadays, the challenges related to technological and environmental development are becoming increasingly complex. Among the environmentally significant issues, wildfires pose a serious threat to the global ecosystem. The damages inflicted upon forests are manifold, leading not only to the destruction of terrestrial ecosystems but also to climate changes. Consequently, reducing their impact on both people and nature requires the adoption of effective approaches for prevention, early warning, and well-coordinated interventions. This document presents an analysis of the evolution of various technologies used in the detection, monitoring, and prevention of forest fires from past years to the present. It highlights the strengths, limitations, and future developments in this field. Forest fires have emerged as a critical environmental concern due to their devastating effects on ecosystems and the potential repercussions on the climate. Understanding the evolution of technology in addressing this issue is essential to formulate more effective strategies for mitigating and preventing wildfires.

## 1. Introduction

The destruction of the world’s forests is accelerating precipitously, both due to intensive cultivation and fires. Compared to the beginning of the century, the forest area has halved. Other factors include rising temperatures and drought, in turn fueled by climate change. In 2021 alone, 9 million hectares were lost, an area the size of Portugal (Europe). Of these, 7.8 million were lost by Russia, Canada and the United States. In Italy (Europe-Mediterranean area), 723,924 hectares went up in smoke in the previous 14 years, while in 2021 alone as many as 159,437 forested hectares were devastated by flames, a borderline condition for a world that year after year is increasingly intensively exploited [1].

Forest fires represent a grave and multifaceted threat to forests and land ecosystems, necessitating immediate attention and proactive measures. These fires have substantial ecological, economic, and social ramifications, resulting in extensive damage to forested areas and beyond.

The causes of forest fires encompass both natural and human-induced factors. Natural causes include lightning strikes, volcanic activity, and spontaneous combustion. Human-induced causes involve uncontrolled agricultural practices, land-use changes, arson, and negligence. Additionally, climate change contributes to the escalation of fire conditions, intensifying the occurrence and severity of wildfires [2].

Ecologically, forest fires have profound consequences. They result in the loss of biodiversity by destroying habitats and endangering plant and animal species. Soil degradation is another significant impact, leading to erosion, nutrient loss, and reduced fertility. Furthermore, forest fires disrupt crucial ecological processes, such as nutrient cycling, and alter patterns of succession and regeneration. These fires also increase the vulnerability of ecosystems to invasive species [3].

The impacts of forest fires extend to land ecosystems. Deforestation and habitat loss are irreparable damages caused by fires, with severe implications for ecosystem integrity. Forest fires release substantial amounts of carbon emissions, exacerbating climate change and global warming [4]. Water resources and hydrological cycles are also affected, leading to water quality issues, reduced availability, and disruptions to watershed functions. Socioeconomically, forest fires result in the loss of livelihoods, displacement of communities, and significant economic burdens on affected regions.

To address this threat effectively, comprehensive fire management strategies are essential. Early detection and rapid response systems play a crucial role in monitoring and surveillance, enabling timely action. Fire prevention measures [5], including controlled burns, fuel management, and public awareness campaigns, are crucial for reducing fire risks. Fire suppression techniques involve the deployment of firefighting resources and infrastructure to contain and extinguish fires. Active community involvement and capacity building enhance fire-safe practices and community resilience. International cooperation and knowledge sharing facilitate collaborative approaches to fire management, leveraging shared expertise and resources.

The theoretical background of wildfire detection and monitoring involves multiple scientific disciplines and principles. Key elements include fire behavior and spread, remote sensing, data analysis, and modeling.

Understanding fire behavior and spread is crucial for effective detection and monitoring. This involves studying the physical properties of fire, such as heat transfer, combustion processes, and fire dynamics. By comprehending how fires ignite, spread, and interact with the environment, scientists and practitioners can develop more accurate detection and monitoring strategies.

Remote sensing plays a significant role in wildfire detection and monitoring. It involves the use of satellite imagery, aerial photography, and other sensor technologies to capture data about fire occurrences, smoke plumes, and burned areas. Remote sensing enables the identification and tracking of wildfires over large geographic areas, providing valuable information for decision-making and resource allocation.

Data analysis is a vital component of wildfire detection and monitoring. It involves processing and interpreting data collected from various sources, such as satellite imagery, weather stations, and ground-based sensors [6]. Data analysis techniques, including image processing, statistical analysis, and machine learning, help identify fire signatures, detect anomalies, and provide timely information for fire management.

Modeling is another crucial aspect of wildfire detection and monitoring. Mathematical and computational models [7] are used to simulate fire behavior, predict fire spread, and assess the potential impacts of wildfires. These models incorporate factors such as weather conditions, fuel characteristics, and terrain to generate predictions and inform decision-making.

The theoretical background of wildfire detection and monitoring encompasses an understanding of fire behavior, the utilization of remote sensing technologies, data analysis techniques, and the development of models to predict and monitor wildfires. By integrating these theoretical foundations, researchers and practitioners can enhance the effectiveness of wildfire detection and monitoring systems, leading to more efficient fire management and mitigation efforts.

By integrating advanced ICT systems seamlessly into an environment, to the extent that these highly technological systems become an intrinsic part of it, we can enhance the environment with additional features. These features primarily include self-monitoring and self-protection capabilities, granting the environment a basic level of intelligence. This intelligence allows the environment to operate not only reactively but also proactively, prioritizing its self-protection. Consequently, the environment evolves into an intelligent environment or, more precisely, an intelligent self-monitoring, self-protecting, and self-aware environment. It responds to changes promptly and alerts the responsible humans in real time, enabling them to take appropriate actions to prevent further degradation. In [8], Stipanicev, D. et al. provide an overview of the architecture of such an intelligent environment, which is based on an advanced sensor network known as the observer network. Furthermore, it delves into the system architecture of a forest fire monitoring system as an illustrative example. The discussed approach is based on ideas of formal observer introduced in 1987 by Bennet et al. in [9]. They introduced an approach to a study of perception that attempted to be both rigorous and general. The mathematical model has been applied in several scenarios as a part of forest fire detection systems and intelligent forest fire monitoring systems.

However, several challenges must be addressed. Climate change implications and the resultant escalation of fire risks necessitate adaptive strategies. Balancing conservation goals with socioeconomic needs requires careful consideration and integrated approaches. Incorporating traditional ecological knowledge and indigenous fire management practices can provide valuable insights and contribute to more effective strategies. Advancements in technology, such as early warning systems and innovative fire suppression techniques, offer promising avenues for improvement. Strengthening international cooperation and information exchange is vital to address the global nature of this threat.

Moreover, the significant threat of forest fires to forests and land ecosystems demands immediate action. By implementing proactive measures, including sustainable forest management practices, effective fire prevention strategies, and robust policies, we can mitigate the devastating impacts of forest fires. The urgency for global collaboration and concerted efforts cannot be overstated to safeguard our precious forests and land ecosystems from this pervasive threat.

Therefore, prevention plays a crucial role in minimizing the occurrence and severity of wildfires, offering a proactive and sustainable approach. It not only saves lives, resources, and ecosystems but also safeguards the well-being and livelihoods of fire-prone communities. Fuel management for the prevention of wildfires in Southern Europe is often economically unsustainable. Ascoli et al. [5] examines fuel management initiatives in Southern EU countries and proposes sustainable solutions. Innovative initiatives involve both public and private resources to enhance the value of fuel management products and services. Sousa et al. [10] conducted a study on wildfire propagation in Portugal, analyzing various factors such as vegetation, climate, topography, and human influence. They utilized spatial cluster analysis to identify homogeneous regions and employed regression models to understand the contribution of different elements in extensive fire spread. The study revealed spatial variability in the impact of structural factors on fire propagation. A study by Granville et al. [11] analyzed fire growth distributions in Ontario’s Crown forest areas from 1976 to 2019. Industrial forestry operations in Ontario, Canada are limited to reduce the risk of wildfires through the Modified Industrial Operations Protocol (MIOP). They found iterative improvements in fire growth response over time, indicating the effectiveness of MIOP in reducing negative impacts. MIOP allows for operational flexibility while promoting safe practices in industrial forestry operations, with the aim of minimizing the negative impact of fires caused by industrial activities. Athanasiou et al. [12] describe the initial phase of a two-year pilot project on prescribed burning (PB) in Greece. The objective is to reintroduce controlled fire as a reliable and effective tool for wildfire prevention. The project involves planned field experiments to gather knowledge on fire behavior, and its impact on soil, trees, and plant biodiversity. These experimental fires also provide training opportunities for firefighters, land managers, and researchers. Other interesting topics such as examining fire use, fire permits, and safe burning practices are covered in the study by McGee et al. [13], in particular among rural residents in the Edson forest area of Alberta, Canada. The related survey revealed that while most respondents used fire on their properties and were aware of the local fire risk, there was limited recognition of the contribution of agricultural fire use to wildfires.

Additional works have dealt with similar issues located in different parts of the world, such as Australia, New Zealand, and Africa [14]. In [15], the authors show a mathematical treatment of backing fire, while in [16] an operational methodology for directing and influencing the natural direction of a fire is discussed.

Forest fire monitoring, detection, and prevention utilize a range of methodologies, systems, and sensors to enhance early detection, response, and management of wildfires. Remote sensing techniques, such as satellites and aerial platforms, provide real-time data on fire hotspots, smoke plumes, and burned areas. Geographic Information Systems (GIS) integrate spatial data for risk mapping and resource allocation. Weather monitoring systems and prediction models aid in fire weather forecasts and early warning systems. Fire detection systems, including ground-based and satellite-based sensors, identify heat signatures, smoke, and flames for prompt response. Sensor networks continuously monitor environmental conditions, while machine learning and artificial intelligence analyze data for fire detection algorithms. Controlled burning and fuel management practices mitigate fire risk. Community engagement and public awareness programs promote fire-safe practices and early reporting of fire hazards. By implementing these approaches, forest fire monitoring, detection, and prevention efforts can be more effective, reducing the impact of wildfires on ecosystems and communities.

This paper aims to go into detail about the methodologies and major systems currently employed supported by a wide range of sensor technology. Image analysis is a widespread procedure that can employ either static acquisition systems installed on control towers or images from satellites, as is described in [17] by R. Shanmuga et al. More recent methodologies involve the use of drones that allow for faster analysis than satellites, and have lower costs than other technologies being fielded. For example, Zhentian Jiao et al. [18] propose the use of YOLO neural network that is applied to data read from a drone-mounted depth camera, which provides in real time images related to a possible forest fire detention.

Scientific research is making numerous strides to try to counter these trends at a minimum, proposing various alternatives that, from the early 2000s to the present, present various changes and innovations. In fact, today, attempts are being made, for example, with the use of advanced tools on image analysis, through the use of artificial intelligence techniques, applied to the basis of each frame in a video that allow the extraction of each and every detail that can be traced back to a fire principle. Dongqing Shen et al. explore the application of the YOLO neural network in this context [19]. Although the references for these papers are not included in the current bibliography, they provide valuable insights into the topic.

In addition, Barmpoutis et al. [20] describe how the use of wireless sensors of this type can be used in conjunction with cameras to ascertain an actual fire presence and identify its location. Newer communication networks allow intra-node and node-to-gateway communication, thickening the coverage and reaching even those areas that are difficult to reach by radio signal.

The analysis explores the strengths and advantages offered by different technologies employed in fire detection and prevention, such as advanced image analysis, the utilization of artificial intelligence techniques, and the integration of sensor networks. These technological advancements enable the timely detection of fire outbreaks, enhancing early warning systems and allowing for proactive measures to be taken. However, it is important to acknowledge the limitations of these technologies, such as the need for continuous monitoring, potential false alarms, and the challenges associated with their implementation in diverse geographical contexts.

Furthermore, the document examines the potential future developments in the field of fire detection and prevention. It explores emerging technologies and approaches that could further enhance the effectiveness of fire management strategies. The objective is to provide a comprehensive overview of the current state of technology and inspire further research and innovation in the field, as shown in Figure 1. By identifying the strengths, limitations, and future prospects of existing technologies, this analysis contributes to the ongoing efforts to develop more efficient and sustainable approaches for mitigating the impact of forest fires on both the environment and human well-being. The remainder of the document is structured as follows: Section 2 will discuss wireless sensor networks for fire detection. Section 3 will cover the use of different video techniques for fire detection. Section 4 addresses the role of machine learning and artificial intelligence in fire detection, analysis, and prevention. Finally, Section 5 outlines the conclusions and future developments.

## 2. Wireless Sensor Networks for Fire Detection

Nowadays, there is a growing focus on developing an affordable and real-time method for early fire detection using wireless sensor networks (WSN). A WSN consists of multiple nodes, each with several features. The diversity of WSNs lies in their network topology, communication approaches, sensor types, and data processing techniques. By incorporating different type of sensors, wireless nodes have the capability to detect a range of physical parameters including temperature, pressure, and humidity, as well as chemical parameters such as carbon monoxide, carbon dioxide, and nitrogen dioxide. The adoption of this approach enables quicker fire detection compared to conventional methods such as satellite imagery, which involve lengthier acquisition and processing durations. Sensor networks offer an additional benefit over satellite images, as the latter can face limitations in accuracy under specific conditions (e.g., cloudy weather), along with extended scanning intervals and lower resolutions in certain satellites. Enhanced detection capability enables timely intervention before the fire escalates beyond control. Through the integration of Wireless Sensor Networks (WSN), Machine Learning (ML), and Artificial Intelligence (AI) methodologies, it becomes feasible to anticipate potential fire patterns, such as leveraging sensor data such as wind direction, enabling predictive analysis. Ensuring the energy autonomy of devices poses a significant challenge for such networks, especially when strategically situating wireless nodes in forested regions. In addition to contending with limited energy resources, sensor networks are vulnerable to adverse environmental conditions, demanding meticulous deliberation and effective mitigation strategies. Yu et al. [21] propose a network of nodes densely distributed within the forest that collect measured data, such as temperature and relative humidity. They send them to cluster nodes that process data by building a neural network. The network takes the measured data as input and produces a “weather index”, which measures the probability of a fire caused by the weather. In some emergency situations, nodes can detect smoke or abnormal temperature and then send a certain type of alarm to the node manager. The measured data are relative humidity, temperature, smoke, and wind speed. This determines the rate of forest fire hazard. They propose a network composed of a large number of small and economic nodes, with the advantage of obtaining information in fast times and precise forecasts. Figure 2a illustrates a design example of a Wireless Sensor Network (WSN) composed of nodes equipped with integrated microsensors as adopted by the authors in [22]. These nodes are distributed throughout the monitoring area. The purpose of this network is to collect real-time dynamic information such as temperature, humidity, and atmospheric pressure. The collected data are transmitted to routers within the network, which then create a local database and send the information over the Internet. Within the WSN, there are two types of nodes: coordination (COORD) and relegated function (RFD). The ZigBee protocol is employed for wireless communication, allowing data transmission while also enabling a dormant state to significantly reduce energy consumption.

It is crucial to note that accurate geolocation plays a vital role in this setup. The authors stress the significance of precise positioning, as it ensures reliable measurements and facilitates informed decision-making. They propose localization algorithms such as Received Signal Strength Indicator (RSSI), Time-of-arrival (ToA), Angle of arrival (AoA), and Time-difference-of-arrival (TDoA) to achieve accurate geolocation. Aslan et al. [23] present compelling ideas concerning the architecture of wireless sensor networks (WSNs), including sensor deployment schemes, clustering, and communication protocols. The authors’ primary objective is to detect potential fire threats at the earliest possible stage while considering the energy consumption of sensor nodes and the impact of environmental conditions on network reliability. In their article, they develop a noteworthy simulator to validate and evaluate the proposed network architecture. Through simulations, the authors successfully demonstrate the achievement of several objectives, namely, energy efficiency, early detection and accurate localization of fire threats, forecast capability, and adaptability to harsh environments. Furthermore, the authors offer valuable insights into the organization of sensor nodes, emphasizing its influence on system design and performance. They suggest that careful consideration should be given to factors such as the average distance between sensor nodes and the distribution model to enable efficient communication among them. For instance, the choice of layout, such as a square (refer to Figure 3a) versus a hexagonal layout, impacts the number of sensor nodes per cluster. The square layout, with fewer nodes per cluster, effectively manages congestion and enhances system robustness.

Lloret et al. [24] propose an alternative network design similar to the one depicted in the Figure 2b, that introduces a novel network topology capable of determining the required number of devices for covering a specific area. An example of this topology is outlined in Figure 3b, where the number of sensors is much greater than Access Points (APs). This design offers the benefit of system scalability. One notable innovation they present is the utilization of wireless IP cameras in conjunction with multi-sensor systems, interconnected through the IEEE 802.11 g standard. By employing this approach, they are able to detect the presence of fire. Initially, various sensors trigger an alarm, and subsequently, the IP cameras are activated in the relevant area to verify the occurrence of an actual fire. This method helps to eliminate false alarms.

Bayo et al. [25] propose a system designed to perform different measurements at different tree heights, depending on the forest relief. Thanks to this method, they can also detect underground fires and understand how the fire affects the ground cover, the stems, and the tops of the trees. With low energy consumption, the prototype node enables the measurement of internal and external temperature, relative humidity, barometric pressure, and light intensity. The utilization of algorithms, as demonstrated in [26], presents another avenue for enhancing WSNs. The authors emphasize the significance of incorporating intelligent decision-making (IDM) capabilities within the network architecture. By leveraging IDM and predefined sensitivity levels, they successfully activate the required actions and effectively reduce energy consumption. The WSN collects data for use as raw input data, which is transmitted into the control system. A Fuzzy Logic algorithm is developed using parameters such as temperature, smoke, light, humidity and distance. Krüll et al. [27] introduce a comprehensive approach that integrates various detection systems, considering factors such as fire risk, area size, and human presence. This approach is accompanied by a suitable logistics infrastructure, simulation training, and advanced extinguishing technology. In order to prevent false alarms, a remotely operated unmanned aerial vehicle (UAV) equipped with gas sensors and a thermal imaging camera is employed to monitor potential fires. To enhance monitoring effectiveness, various equipment is installed on the airship, including a microwave radiometer (for detecting hot spots), gas and smoke sensors, and a thermal imaging camera. Once fire suppression is carried out, the unmanned airship acts as a firestop, minimizing the risk of fire re-ignition. Cui et al. [28] present an IoT-based network, depicted in Figure 2d, which employs Deep Learning algorithms such as Convolution Neural Network (CNN) for forest monitoring and anomaly detection, yielding favorable outcomes. In the surveillance network, every IoT device establishes communication through 4G internet connectivity. Various types of sensors serve as monitoring devices, measuring variables such as temperature, atmospheric pressure, humidity, and the presence of pollutants such as CO and CO2. One specific approach, outlined in [29], involves the utilization of animals equipped with sensors known as mobile biological sensors (MBS). These sensors, including thermal and radiation sensors with GPS functionality, are attached to the animals. They transmit the MBS’s location, enabling a central computer system to classify the animals’ behavior. The system, depicted in Figure 2c, facilitates the detection of sudden group movements (panic) among the animals through an animal behavior classification method (ABC). Additionally, it allows for the identification of instantaneous temperature changes (thermal detection-TD).

Sahin et al. [30] proposes a study related to the use of a radio acoustic noise system for creating heat maps of forest areas with potential fire risk. The main property of Radio-Acoustic Sounding System (RASS) is linked to the great sensitivity to temperature variations and the possibility of remotely managing the variations in air temperature better than any other surveying instrument in research. In addition, it is capable of continuously tracking intervals at simultaneous multiples with spatial and temporal resolution more precisely, more efficiently, and more cost-effectivelythan any other solution proposed by a network of static sensors. Volumetric Acoustic Scanner (VAS) is a tool that allows to detect an object by acquiring sound waves. In the case of wildfires, Domingos X. Vegas et al. [31] analyze a system that allows to detect the noise emitted by the fire, analyzed in a frequency range between 200 and 500 Hz. In order to be able to detect and locate the fire, it uses multiple microphones, which through the beamforming process are able to output the location of the source through the sum of the various signals picked up by the series of microphones used in testing. These signals, to ascertain the effective holding of a fire, are processed by a neural network so as to have higher accuracy.

Among other remote sensing techniques, that of Light Detection and Ranging (LiDAR) is a possible solution to be able to build a map for the analysis of the wooded area of interest, with a focus on tree volume estimation, habitat characterization, and forest fire estimation. In particular, LiDAR through a laser pulse can determine the distance to an object or surface. Therefore, it can be used to create possible fire scenarios from a map of the wooded area of interest. Marta Fernandez-Alvarez et al. [32] propose a methodology based on UAV LiDAR to characterize forest fuels in a wildland–urban interface (WUI). This kind of work in any case can be extended to forests as well.

A further alternative could be a network of sensors connected to the optical fiber. The fiber optic sensor network (FOSN) is an improvement over traditional sensor networks in that it can be used exactly like a traditional sensor network, but with the advantages associated with fiber optics: little electromagnetic interference, and greater efficiency in terms of signal propagation. Wide bandwidths and low transmission losses are equally crucial, as is the coverage of a large geographical area and the great difficulty in intercepting data during transmission versus radio propagation in metallic cables. The fiber interconnections allow the insertion of the sensors inside the structures to be monitored, without power supply local outside the terminal nodes, reducing the risk of sparks in flammable environments. Montserrat Fernandez-Vallejo et al. [33] analyze the advantages and disadvantages of using these networks of sensors, also providing some examples of use over large distances.

## 3. Video-Based Fire Detection: Enhancing Fire Detection Systems with Visual Analysis

The detection of fires in videos using both the visible and (IR) spectrum is a powerful approach that enhances the capabilities of fire detection systems. By analyzing video footage captured in both spectra, advanced computer vision algorithms can effectively identify and alert authorities to the presence of fires in real-time. Video-based fire detection using the visible and IR spectrum leverages the distinct characteristics of flames, smoke, and heat emitted by fires. The visible spectrum captures the visual cues of flames and smoke, while the IR spectrum detects the thermal signatures and temperature anomalies associated with fire events.

### 3.1. Video Fire Detection in the Visible Spectrum

Extensive research has been conducted over the years on the detection of fire and smoke from video footage within the visual spectral range. A significant contribution to this field was made by Healey et al. [34], who were among the early pioneers to introduce an automated system for real-time fire detection using color video input. Their approach utilized the spectral, spatial, and temporal properties of fire to achieve accurate results. The algorithm developed by Healey et al. can be divided into four main components. Initially, the algorithm divides the image into a grid structure for efficient analysis. It then identifies the rectangles within the grid that have the potential to represent fires. Subsequently, the algorithm labels the connected fire components and finally interprets the labeled components. Based on the interpretation, a fire alarm is triggered if necessary. In [35], Borges et al. propose a method adaptable not only to fire disaster detection, but also to the retrieval of fire-related information from news content. Their approach involves analyzing frame-by-frame changes in specific features within potential fire regions. The proposed method aims to identify and extract relevant fire-related information from newscasts. By analyzing consecutive frames, the algorithm focuses on observing and measuring changes in predetermined features within potential fire regions. This analysis helps identify and track fire-related regions or events represented in news content. Similar approaches were presented a few years later with important improvements. In 2003, Chen et al. [36] introduced a fire detection method that employed video processing and utilized the RGB color model. Their approach involved extracting fire-pixels and analyzing various dynamic features associated with fire from visual images. Based on the saturation of the red component (R), the system determined the presence of fire. Additionally, a fire alarm was triggered depending on the number of fire-pixels and the satisfaction of a specified threshold. Furthermore, Chen et al. [37] extended their research to detect smoke in a separate work. They examined clutter within visual images to extract both fire and smoke pixels. By leveraging the RGB color model, they identified specific fire features such as color, area size, surface characteristics, boundary roughness, and asymmetry of the fire region. In a related study, Borges et al. [38] also employed the RGB color model to extract fire features. These features included color properties, area size, surface characteristics, and boundaries roughness. Additionally, the asymmetry of the fire region was taken into account. By utilizing these features, Borges et al. aimed to develop a probabilistic framework for fire detection. The works by Chen et al. and Borges et al. demonstrate the utilization of the RGB color model and video processing techniques for fire detection. By analyzing various visual features and employing decision-making mechanisms, these methods offer potential solutions for early fire detection and the identification of smoke. Such advancements contribute to enhancing fire safety measures and prompt response strategies in critical situations. Marbach et al. [39] conducted research on fire detection using a technique based on the temporal intensity variation of the fire. They employed the "YUV" representation of the video, diverging from previous studies that utilized the RGB model. The system they developed extracted various fire features, such as luminance of fire pixels, the count of active pixels, and the number of saturated pixels, from a designated fire region. This region was identified through a temporal accumulation of time derivative images.

In addition, Töreyin et al. [40] presented an approach based on YUV color space, while in [41], Rossi et al. employed both the RGB and YUV color spaces in their fire detection approach. By utilizing both color spaces, they aimed to discriminate fire elements that are located far from the identified fire region in large areas and they applied K-means clustering technique specifically to “V” channel in order to identify the most interesting areas corresponding to fire, as fire tends to exhibit distinct characteristics in this color axis. Similarly, Rudz et al. [42] and Celik et al. [43] adopted the “YCbCr” color space in their work, which presented a fire detection method applicable to a generic color mode. Fire-pixels were categorized based on their chrominance, made possible by utilizing YCbCr. Unlike RGB, YCbCr allows for the separation of luminance and chrominance, as depicted in Figure 1 and Figure 3 shown in [43], thereby enabling effective classification of fire pixels. Instead, Celik et al. [44] also proposed an algorithm that can be used in parallel with other fire detection systems or as a stand-alone system, using a new color model in CIE L*a*b* color space to identify fire pixels. Specifically, the “*a*” color channel represents colors ranging from red to green, with positive and negative values indicating red and green, respectively. Yuan et al. [45] employed the “Lab” color mode, focusing on the use of the “*a*“ channel within this color model to extract chromatic fire features. The authors found that leveraging “*a*” channel yielded highly effective results for fire segmentation, making it a suitable choice for their study. In [46], a segmentation problem is faced and computer vision-based fire detection algorithms are presented, where both Lab and YCbCr color spaces are adopted, along with K-means clustering algorithms. Other color spaces have been used in fire detection: HSV/HSL color space (Hue, Saturation, Value also known as Hue, Saturation, Brigthness HSB, and Hue, Saturation, Lightness, respectively) has been presented in techniques for fire detection such as in [47,48,49,50], and in [51] used alongside with RGB color model. In their work, Chmelar et al. [52] presented an approach using both RGB and HSV color model, eventually illustrating why the HSV model was more suitable for their method. CIE L*u*v* color space has been adopted by Pritam et al. [53] in 2017, with various thresholds used to differentiate fire in any fire frame. HSI (Hue, Saturation, Intensity) is instead adopted by Horng et al. [54], where fire flames features are extracted by 70 flame images, and in [37] by Chen et al. with RGB color space. Starting from RGB color model, Khatami et al. [55] in 2015 presented a new color space, specifically designed for fire-involving application: Fire-based Color Space (FCS) has been obtained with a computational search method based on K-medoids clustering. Their approach uses a conversion from RGB color system to FCS color system through a 3 × 3 matrix.

Töreyin and Dedeoĝlu and their research group introduced an alternative approach that relies on wavelet analysis of videos instead of utilizing color models, as documented in several works [56,57,58,59,60]. In their initial work [56], they combined traditional motion and color analysis with wavelet domain analysis. Specifically, they employed spatial and temporal wavelet transforms to detect flame and fire flickers, which represent irregular features within the fire region. Subsequently, all gathered information was utilized to make informed decisions. The same methodology was applied in [57] for videos captured by ordinary cameras and in [58] for infrared videos. In both [59,61], a weak classifier is introduced that utilizes both temporal and spatial information of flames. Additionally, these works incorporate the use of Hidden Markov Models (HMM) to represent flames and other moving objects that share the same color as fire. Similarly, Teng et al. [62] also employed HMMs for fire pixel detection, encompassing the detection of moving pixels, fire-color inspection, and pixel clustering. Results for each step of Teng et al. [62] proposed method show input video sequences, the results after being processed by moving pixels detection, and the results of fire-color detection. Another feature is the clustering results, whereas the last row is the final results after using HMM application. An innovative method has been introduced in [63] for fire detection in videos, employing Least Mean Square (LMS)-Based Active Learning. This approach is used alongside the conventional techniques such as Hidden Markov Models (HMM), background subtraction, wavelet domain analysis, and moving object detection. What sets it apart is the fusion of decisions made at each stage of the algorithm using an adaptive algorithm that updates weights through the least mean square method during the training stage. Ko et al. introduced an additional classifier in conjunction with wavelet domain analysis, as described in [64]. Their approach encompasses fire region detection, analysis of the luminance map within the fire region to eliminate non-fire pixels, and the creation of a temporal fire model using wavelet coefficients. This model is then applied to a two-class Support Vector Machines (SVM) classifier. Ultimately, the SVM classifier serves the purpose of verifying the identified fire pixels. Likewise, Habiboglu et al. [65] and Dimitropoulos et al. [66] utilize the SVM classifier to train their algorithm, which is based on the fire spatio-temporal covariance matrix. On the other hand, Dimitropoulos et al. employ the SVM classifier for the classification of the fire region.

In 2018, Wu et al. [67] introduced a fire smoke detection system that relies on the Robust AdaBoost (RAB) classifier. Their goal was to enhance training and classification accuracy within the system. Background subtraction is a widely employed technique for video fire detection. Xu et al. utilized this method in conjunction with the median filter algorithm to perform moving target detection in grayscale images [60]. Verstock et al. also employed background subtraction, specifically for infrared images [68]. Similarly, Dimitropoulos et al. used background subtraction to define fire regions in their work [66]. In 1996, Foo introduced a video fire detection system for aircraft that relied on statistical features such as median, standard deviation, and first-order momentum obtained through histogram analysis. Additionally, the system utilized image subtraction of two consecutive frames to extract relevant features [69]. Following this pioneering work, other studies began adopting a statistical approach for their proposed fire detection systems. Phillips et al. [70] were the first to present a solution specifically designed for non-stationary cameras. Their method involved using a Gaussian-smoothed color histogram to detect fire-color pixels. Building upon this statistical framework, Yuan et al. [45] implemented a fire detection and tracking algorithm based on median filtering. Similarly, for infrared images, a comparable approach utilizing median filtering has been employed [68,71].

Neural networks have found extensive application in video fire detection. Xu et al. were among the pioneers in utilizing neural networks, specifically Back Propagation artificial neural networks, as described in their work [60]. Bayesian networks have also been employed in the context of video fire detection in studies conducted by Borges et al. [38] and Ko et al. [72]. Muhammad et al. explored the use of computationally efficient Convolutional Neural Networks (CNNs) for video surveillance applications in both 2018 and 2019 [73,74]. They harnessed the power of CNNs to achieve efficient video fire detection. Kim et al. adopted a combination of Region-based CNN (R-CNN) and Long-Short Term Memory (LSTM) in their work [75]. This approach allowed them to determine the presence or absence of fire within short time intervals. In their research, Kolesov et al. [76] propose an innovative method for fire and smoke detection by utilizing Optimal Mass Transport (OMT) optical flow. They approach the detection process as a supervised Bayesian classification problem, incorporating spatio-temporal neighborhoods of pixels. The feature vectors used in this classification consist of OMT velocities, along with the R, G, and B color channels. Smoke detection is an equally crucial aspect alongside fire detection, warranting a brief overview. In the study conducted by Yuan et al. [77], the primary objective was to reduce false positives, and their approach revolved around an accumulative motion model that utilized integral images. The process involved integrating the images, estimating motion orientation and acceleration using velocity vectors, and finally utilizing motion accumulation to estimate smoke orientation as shown in Figure 7 of [77]. Chen et al. also contributed to the field of smoke detection with their works [37,78]. Their smoke detection system employed two decision criteria for smoke-pixel determination based on chromaticity and the diffusion characteristics of smoke. Smoke detection is also accomplished by modeling the smoke using a Mixture of Gaussians (MoG) technique [79], which analyzes the energy variation in the wavelet domain. By employing this approach, the system considers the changes in image energy caused by external factors such as changes in luminance or the presence of smoke. Table 1 summarizes the different features analyzed by the studies examined for smoke detection.

### 3.2. Video Fire Detection in the Infrared Spectrum

IR (infra-red) cameras have the unique capability of capturing the thermal radiation emitted by objects or groups of objects in the surrounding environment. This characteristic makes them equally valuable and effective during both daytime and nighttime scenarios, as they can detect and identify thermal sources of specific significance. This capability enables the capture of data across a wide range of resolutions, accommodating various fire detection scenarios. In their research, Chi Yuan et al. [71] propose employing IR cameras mounted on Unmanned Aerial Vehicles (UAVs) as shown in Figure 4.

This approach allows for the differentiation of fire zones, characterized by a greater concentration of bright pixels, from moving objects (which also exhibit a high proportion of luminous pixels), such as animals. A similar framework can be incorporated into a satellite-based image analysis system, enabling a comparison of results to confirm the presence of a forest fire. Meenu Ajith et al. [80] propose an algorithm consisting of two phases: extraction and segmentation of the focal area.

In the first phase, various attributes of each pixel within a given sequence of images are analyzed, including brightness and motion fields. These attributes are extracted to determine the luminosity and velocity range. In the second phase, known as segmentation, a cluster of pixels with similar characteristics is outlined, allowing for the classification of the object as either fire or non-fire. Utilizing IR cameras presents a favorable solution for fire detection and localization. The technique’s advantages stem from its ability to identify highly luminous pixels against a dark background, enabling the distinction between fires and other objects, such as animals, through suitable algorithms. However, there are certain drawbacks associated with this approach. One primary limitation is the inability to detect and classify clouds. Consequently, it becomes challenging to differentiate a cloud with a dense concentration of smoke, indicating a fire, from a typical hydrometeor.

### 3.3. Satellite

For several years, the analysis of satellite images has been a widely used method, providing a comprehensive overview of areas of interest for potential fire detection. It is important to note that not all satellites are specifically designed for Earth observation and environmental monitoring.

Satellites can be categorized based on their orbit. One category is GEO (geostationary orbit), situated at an altitude of 35,786 km and characterized by zero inclination. This orbit allows the satellite to remain relatively stationary with respect to the Earth’s surface, providing a constant view of the same area. While GEO satellites offer high temporal resolution, their spatial resolution is relatively low.

Another category is LEO (Low Earth Orbit), which orbits at an altitude of approximately 2000 km, and MEO (Medium Earth Orbit), which orbits at altitudes up to 11,000 km. Compared to GEO satellites, LEO and MEO satellites provide higher temporal resolution but lower spatial resolution [20]. Due to the significant orbital travel time, GEOs are not particularly suitable for real-time fire monitoring. However, they can be utilized retrospectively to estimate the extent of burned area, making them valuable for post-fire analysis rather than fire detection and prevention studies. In the case of sun-synchronous satellites (i.e., LEO and MEO), extensive research has been conducted on the analysis of the images they capture. Among the multispectral image sensors used, the focus lies on AVHRR (Advanced Very-High-Resolution Radiometer). AVHRR offers six channels, including three thermal infrared channels, with a spatial resolution of 1 km. These images measure Earth’s reflectance and are commonly used for global cloud cover monitoring. The study by Zhanqing et al. [81] examines the significance of cloud density in fire detection. By utilizing a neural network and analyzing NOAA-14 satellite images, the research distinguishes smoke resulting from fires from both clouds and the Earth’s surface based on reflectance properties. This approach sheds light on the evaluation of cloud density for accurate fire detection.

The MODIS (Moderate Resolution Imaging Spectroradiometer) sensor operates across 36 spectral bands, covering wavelengths from 0.4 to 14.4 μm, with varying spatial resolutions. It includes 2 bands at 250 m, 5 bands at 500 m and 29 bands at 1km resolution. This advanced technology has been extensively utilized in various research studies. One such study conducted by Wilfrid Schroeder et al. [82] exploits imagery captured by the Landsat-8 satellite, which carries the Operational Land Imager (OLI) and the Infrared Thermal Sensor (TIRS). The OLI is a push-broom sensor equipped with nine spectral channels, providing a spatial resolution of 30 m. The research conducted by Schroeder et al. leverages these satellite images to examine and analyze specific aspects of interest. Figure 1 of [82] shows the 4500 fire-affected pixels (marked in red). The peak in the radiance distribution coincides with the nominal saturation radiance (24.3 W/(m^2^ sr μm)). Pixels exceeding the nominal saturation are representative of analog high saturation.

The evaluation process relies on an active radiometric signal focusing on a region that exhibits an unusual change in reflectance compared to the background. Additionally, it takes into account the concentration of a window of pixels marked as red by the infrared sensor. This approach enables precise localization of the fire area with a low margin of error. In another study, Csizar et al. [83] propose a fire detection method that combines ASTER (Advanced Spaceborne Thermal Emission and Reflection Radiometer) and MODIS. By adapting the fire mask proposed by MODIS with ASTER’s broader coverage, the method achieves improved accuracy in detecting fire areas in Northern Eurasia. This approach allows for the inclusion of a larger area, where there is a slightly greater concentration of focus pixels exhibiting anomalous reflectance variations. Following MODIS, VIIRS (Visible Infrared Imaging Radiometer) was deployed on meteorological satellites such as NOAA-20. VIIRS offers 22 spectral bands, including 16 moderate-resolution bands (M bands, 750 m), 5 imaging resolution bands (I bands, 375 m), and 1 panchromatic day/night band (750 m). However, relying solely on satellite image analysis techniques without integrating ground-based instruments may not be sufficient for real-time fire detection. Despite the relatively close proximity of sun-synchronous satellites to the Earth, which allows for faster coverage of the planet’s orbit, there is still a significant time delay in timely fire detection. In this context, the speed of detection becomes a crucial factor, despite the higher precision compared to other ground-based instruments. Table 2 shows the characteristics of the sensors in terms of spatial resolution, spectral band, advantages and disadvantages for AVHRR, MODIS, and VIIRS.

## 4. The Role of Traditional Machine Learning and Deep Learning in Fire Detection and Prevention

### 4.1. Detection of Wildfires Using a Machine Learning Approach

Detecting wildfires automatically poses a significant challenge due to the complexity of developing a deterministic algorithm that can accurately determine the occurrence of a wildfire based on a given set of features. However, Machine Learning (ML) proves highly valuable in such scenarios by enabling the utilization of powerful models to extract information and gain insights directly from the data. In the realm of ML-based automatic wildfire detection, there are several techniques employed, with the most prevalent ones being computer vision approaches that utilize both images and videos, as well as classification models trained with environmental data. The subsequent sections will delve into a comprehensive description of these two solutions, elucidating their workings and applications. Training a machine learning model with environmental data such as humidity, temperature, wind speed etc. both in case of a wildfire occurrence and during normal conditions may be a simple and straightforward solution. An early work that proposed this approach is that of Liyang Yu et al. [21] in which authors presented a wildfire prevention and detection system based on clusters of sensors spread across wooded areas. This system implements a simple feed-forward neural network used to assign a real-time forest fire danger rate for each cluster of sensors, on the basis of real-time environmental data collected. This architecture considers the sensors as the input layer of the neural network whereas an hidden layer, whose purpose is to aggregate data to calculate weather indices, is implemented through cluster header nodes, one for each cluster of the sensor network. The output layer of the network is implemented with a manager node which, analyzing weather indices received from cluster nodes, is able to detect anomalies in the data and send an alert whenever a wildfire is starting. Forest fire danger rates can provide valuable insights for implementing preventive measures, as they enable the identification of high-risk areas prone to ignition. A similar approach is presented by Ghorbanzadeh et al. [84], where they utilize a feed-forward neural network to assign a forest fire susceptibility index to various forest regions in northern Iran. This solution aids in assessing the vulnerability of different areas and supports proactive measures against forest fires. In the aforementioned work, the authors have employed environmental data to train their model. They carefully selected 16 variables, encompassing both general factors such as annual temperature, annual rainfall, and altitude, as well as more specific factors such as slope aspect of the region, plan curvature, and distance to the nearest stream/road. This comprehensive selection of variables ensures that various crucial aspects of the environment are taken into account during the model training process. The neural network used to process these variables consists of a 16 neurons input layer, a 28 neurons hidden layer whose optimal number of neurons was obtained through a three-fold cross validation and a final output layer implementing a logistic function whose value ranges from zero to one. According to this model, pixel areas with an output value close to 1 indicate a higher probability of forest fires. By effectively training the neural network, it becomes possible to generate accurate heat maps that highlight areas of greater vulnerability. In another study conducted by Wenyuan Ma et al. [85], the Random Forest algorithm is employed to identify the most significant environmental, social, and economic features from an initial set of 21 variables across different regions of continental China. The research demonstrates that the relative importance of these features varies with different environmental conditions. The study provides valuable insights for wildfire prevention, delivering significant value in understanding the key factors contributing to wildfire occurrences. In conclusion, Zechuan Wu et al. [86] employed environmental data to develop a methodology that utilizes a traditional feed-forward neural network to simulate the advancement of the fire front. They conducted a comparison with the Wang Zhengfei fire physical velocity model integrated with a Cellular Automaton (CA) framework [87], and obtained more accurate results in terms of estimating fire spread. However, there are certain limitations to using environmental data directly for training Machine Learning (ML) models. Firstly, while feeding models with historical data obtained from external archives can facilitate the construction of accurate prediction models, it may not enable the active detection of a wildfire at its onset. On the other hand, employing a model for real-time wildfire detection necessitates the acquisition of data in real-time from remote sensors, imposing significant constraints and requirements on the system to ensure its effectiveness. For effective wildfire detection, sensors must be extensively deployed in the areas of interest, with a high concentration to detect the initiation of wildfires promptly. Additionally, the ML models should efficiently process data from the sensors and provide timely feedback to alert the authorities. Recognizing these challenges, some researchers have begun exploring alternative solutions for wildfire detection. One of these solutions, based on computer vision, will be discussed in the subsequent section.

Sun et al. [88] proposed a development of a forest fire susceptibility model using the LightGBM (an ensemble learning method - short for light gradient-boosting machine) for Nanjing Laoshan National Forest Park, resulting in an accurate fire susceptibility map. Eight variables related to topography, climate, human activity, and vegetation were selected for modeling based on correlation analysis. Logistic regression (LR) and random forest (RF) models were also employed for comparison. Identification of significant factors, such as TMP and NDVI, through importance ranking, providing valuable insights for fire management. The results identified temperature as the primary factor for fire occurrence in the area. Application of LightGBM extends its usage to fire susceptibility prediction, demonstrating its effectiveness.

Zhou et al. [89] present a new approach using an event-response tree-based model to effectively allocate different firefighting resources based on the fire suppression index (SI). This index considers factors such as time, cost, and the impact of deploying resources in suppressing fires. The model aims to improve the efficiency of dispatching resources and control fires in a timely manner. To validate their method, the researchers compared it to the commonly used mixed-integer programming (MIP) model using historical fire data from Nanjing Laoshan National Forest Park. The results demonstrate that the Event Response (E-R) tree-based resource scheduling is equally effective in allocating resources compared to the MIP model. It demonstrates effective resource scheduling and provides clear visibility into the relationship between different resource-dispatching processes. However, the subjective nature of the method, partly based on AHP, suggests the need for further investigation and exploration of alternative methods for assigning weights to factors.

Yang et al. [90] proposed a one-class model for fire detection that focuses on high precision and real-time detection. Their approach directly constructs training samples using fire pixels, without complex feature transformation, and incorporates a batch decision-making strategy to enhance detection speed. On the other hand, on the sub-field of wildfire prevention, Abdollahi et al. [91] employed a Shapley additive explanations (SHAP) model to interpret the results of a deep learning (DL) model for wildfire susceptibility prediction. Their research incorporated various contributing factors and utilized SHAP plots to identify the influential parameters, assess their relative importance, and provide insights into the decision-making process. In addition, Cilli et al. [92] developed an explainable artificial intelligence (XAI) framework for assessing wildfire occurrence in a Mediterranean landscape of Southern Europe. Their study demonstrated the framework’s efficiency and statistical robustness in analyzing wildfire occurrence, highlighting climate as the primary driver. Additionally, the model effectively identified areas where other drivers played significant roles. The study aimed to contribute to the scientific literature on the application of AI in understanding stochastic natural disasters such as wildfires. Bountzouklis et al. [93] developed an explainable artificial intelligence framework to classify the causes of fire ignition in Southern France. Their study successfully predicted the sources of unknown caused wildfires, with natural fires achieving the highest accuracy compared to accidental and arson fires. The analysis identified spatio-temporal properties and topographic characteristics as significant features in determining the classification of unknown caused fires in the region.

### 4.2. Computer Vision and Convolutional Neural Networks (CNN)

The pioneering work of Cappellini et al. [94] has been instrumental in the application of artificial intelligence (AI) for fire detection. Their research has laid a strong foundation for subsequent advancements in utilizing AI algorithms to address the challenges associated with timely and accurate fire detection. Similarly, other notable contributions such as Okayama [95], Arrue et al. [96], and Chen et al. [97] have also made significant contributions in proposing the use of neural networks in combinations with smoke sensor data, infrared-image processing techniques and Fourier Transform Infrared (FT-IR) spectroscopy for gas measurements, in their respective works. Nowadays, the research community has high expectations in the use of artificial intelligence in the field of wildfire detection and many researchers have proposed different algorithms and methodologies to effectively and timely detect wildfire occurrence. Mahdi et al. [98] categorized machine learning approaches for fire detection into two main groups: traditional methods and deep learning methods. Traditional methods encompass classical algorithms such as decision trees and support vector machines (SVM), while deep learning methods represent the prevailing models that include various types of artificial neural networks. Deep learning models possess the ability to automatically select features, resulting in high-performance outcomes. However, they necessitate significant computational power and high-quality datasets for effective training and deployment in real-world scenarios. An illustration of deep learning models in fire detection can be found in the recent work by Seydi et al. [99]. They developed a deep learning framework named Fire-Net, training it with satellite images captured by Landsat-8 in various regions such as Australian and North American forests, the Amazon rainforest, Central Africa, and Chernobyl (Ukraine). Additionally, Abdusalomov et al. [100] utilized a modified version of the Detectron2 platform from Meta AI to achieve highly accurate fire detection. Xue et al. [101] focused on enhancing the well-known YOLOv5 model specifically for fire detection tasks. The literature is replete with system proposals that employ deep learning methods, and this trend appears to be on the rise.

A new trend is emerging in the development of AI and ML models for wildfire detection, as showcased by the latest works published by the research community. This trend emphasizes the utilization of specialized hardware/embedded systems. Such an approach holds great promise, particularly in terms of enhancing the performance of fire detection systems with respect to response time, surpassing the capabilities of architectures designed for general-purpose CPUs. In a recent study conducted by Thangavel et al. [102], the feasibility of employing AI technologies directly on-board satellites for near real-time fire detection was examined and confirmed. The research demonstrated the successful utilization of a combination of specialized hardware, AI on-the-edge paradigms, and hyperspectral imagery. The paper introduced a one-dimensional convolutional neural network (1-D CNN) specifically designed to accommodate on-board implementation and various proposed hardware designs are discussed in the study. The authors considered three hardware accelerators for model implementation: Intel Movidius NCS-2, Nvidia Jetson Nano, and Nvidia Jetson TX2. By employing dedicated on-board hardware for wildfire detection, the response time is significantly reduced, and the system’s efficiency is improved. This approach eliminates the need to transfer hyperspectral imagery to the ground station for processing through AI algorithms. Instead, only the vector data (point or polygon) of the fire, along with the already flagged data intended for the appropriate wildland fire dispatcher based on location, need to be downloaded. George L. James et al. [103] conducted a recent study that shares similarities with the aforementioned research. Their paper presented a system employing transfer learning techniques to develop a neural network architecture suitable embedded systems. To create the model, they started with a pre-trained MobileNetV2 architecture developed by Google and made modifications to tailor it for their requirements, ensuring improved response time and computational efficiency. The authors evaluated the models based on various performance indicators, including flash usage, peak RAM usage, inferencing time, and an overall performance indicator that combines these metrics. These indicators are crucial as the authors aimed to develop a lightweight model for deployment in embedded systems, such as the Arduino Nano 33 BLE Sense board utilized in their study. In their study, G. Peruzzi et al. [104] developed a system prototype that leveraged both audio and video data to detect and alert the presence of wildfires. To achieve better performance, the researchers utilized two convolutional neural networks (CNNs) in combination: one for processing the audio data and another for analyzing the video data. They also employed the MobileNetV2 architecture from Google like the previous paper, as the primary objective of their research, much like the aforementioned studies, was to create a classification model specifically designed for deployment on embedded systems. In addition, the works by Johnston et al. [105], Arguello et al. [106], and Khalifeh et al. [107] are closely associated with the emerging trend of deploying high-speed classification models directly on embedded devices.

Another area of extensive research focuses on the utilization of computer vision models for fire detection. Recent advancements in the development of highly robust and effective models have facilitated real-time solutions that were previously unattainable solely through the use of sensors. Among the various types of machine learning models, Convolutional Neural Networks (CNNs) have emerged as the most promising solution in this domain. Most state-of-the-art frameworks today are based on CNNs or their variants, enabling significantly higher levels of reliability in fire detection compared to traditional sensor-based approaches. Frizzi et al. [108] present a notable work in the field, demonstrating a precise method for detecting smoke emitted by starting wildfires in video captures. They employ a nine-layer CNN that automatically learns relevant features from video frames, eliminating the need for a separate feature extraction phase. Another noteworthy solution proposed by Yichao Cao et al. [109] deals with video recognition of smoke generated by wildfires. The authors suggest using pan-tilt-zoom cameras (PTZ) to cover large woodland areas and propose an intriguing model called Attention Enhanced Bidirectional Long Short-Term Memory Network (ABi-LSTM) as the detection component of the system. In this architecture, a Bidirectional Long Short-Term Memory Network (LSTM) serves as the spatial feature extraction network, while the temporal attention network is employed to extract spatio-temporal features and evaluate different sections of the video frames individually. The extraction of features from raw frames is accomplished using a pre-trained CNN model called InceptionV3, developed by Google. According to the authors, this model is highly accurate and easily deployable. In a more intricate approach, Renjie Xu et al. [110] presented a system that employs multiple detection models simultaneously. The proposed architecture consists of three distinct CNN models: You Only Look Once version 5 (YOLOv5), EfficientDet, and EfficientNet. YOLOv5 and EfficientDet are utilized for the actual object identification function, as they complement each other’s limitations. YOLOv5 excels at detecting large and well-established wildfires but struggles to identify smaller fire zones, potentially missing some of them. On the other hand, EfficientDet is more meticulous in object detection, rarely overlooking potential fire objects, but it is less adept at identifying extensive fire areas. Combining these two models is expected to yield superior overall results. The third model, EfficientNet, functions as a binary classifier, analyzing the entire image to determine whether it contains fire objects or not. The results obtained from object detection and image classification are then forwarded to a decision strategy module responsible for determining whether the identified objects should be considered as fire or discarded. Essentially, the role of the EfficientNet model is to detect false positives and filter them out. This is crucial because the first two models may mistakenly identify certain objects, such as the sun or the colors of a sunset, as fire objects. In summary, the performance of this comprehensive architecture, in terms of precision, average recall, and particularly false positive rate, is well-suited for real-world applications. Compared to other proposed object detection systems, this ensemble learning architecture is more robust in mitigating false alerts. Table 3 compares the two main approaches presented in the previous section, highlighting their advantages and disadvantages.

Forest fires in their early stages tend to have a relatively small size compared to the wide areas covered by deep learning fire detection systems. This size difference poses a challenge for the model as it may fail to capture important information related to these small fires. The existing model might struggle to learn and detect such small targets effectively. To address this problem, Xue et al. [101] introduced an improved model in 2022, which is based on YOLOv5. The proposed method aims to enhance the detection of small forest fire targets. The improvements in the model primarily focus on enhancements to the Backbone layer and Neck layer of YOLOv5, on the replacement of Spatial Pyramid Pooling-Fast-Plus (SPPFP) Module with the Spatial Pyramid Pooling-Fast-Plus (SPPFP) module, the addition of the Convolutional Block Attention Module (CBAM) attention module, and the adaptation of the Path Aggregation Network (PANet) to the Bi-directional Feature Pyramid Network (BiFPN). According to the results obtained, the proposed improvements have demonstrated effective enhancements in the method. The improved model based on YOLOv5 with the modified Backbone layer and the substitution of the SPPF module with the SPPFP module has shown improved performance in detecting small forest fire targets. This suggests that the model is better equipped to learn and recognize crucial information related to early-stage forest fires, despite their small size. Another common problem encountered in forest fire detection is related to the distinct features and morphology of fires and flames. These differences can lead to an increased number of false positives in the detection process and can also hinder the adaptability of the detection system. To address these issues, Chen et al. [111] presented an improved multi-scale forest fire detection model called YOLOv5s-CCAB in their 2023 work. The YOLOv5s-CCAB system incorporates several key components to enhance its performance. Firstly, it includes a Coordinate Attention (CA) module, which likely helps the model focus on relevant fire-related regions by emphasizing spatial coordinates. Additionally, Contextual Transformer (CoT) and CoT3 modules are incorporated to capture contextual information and improve the understanding of complex fire patterns at different scales. Through the use of these enhancements, Chen et al. achieved a high level of detection accuracy and speed with their proposed method. This improved model enables real-time detection of multi-scale forest fires, making it suitable for timely response and intervention in fire incidents. An alternative approach to address the aforementioned problem is described in the research conducted by Qian et al. [112]. Their work presents an algorithm that utilizes weighted fusion to identify forest fire sources in various scenarios. In this approach, two independent weakly supervised models, YOLOv5 and EfficientDet, are employed. These models are trained and make predictions on the datasets simultaneously. The algorithm combines the predictions from these models using a technique called the weighted boxes fusion algorithm (WBF). The WBF algorithm processes the individual prediction results and generates a fusion frame that combines the information from both models.

Among neural networks, one that is quite often used for image analysis is the GNN (graph neural network). It basically consists of a class of neural networks for processing data that can be represented as graphs. In case of computer vision CNNs (convolutional neural networks), these can be seen as a GNN applied to structured graphs such as pixel grids. Mingyang Wang et al. [113], in their work, define a GNN based on the dynamic characteristics of the images. The method converts the input features of the nodes of the graph into different relational features, establishing pairs of nodes representing different points of view of the test images. The dynamic update of the characteristics of the images is done through a future-bank relationship which allows to estimate the similarity of these and improve their recognition rate. Rabah Attia et al. [114] offers a very effective method of image analysis, based on Transformers, which allows a strong saving in computational terms to be able to segment an image, thus obtaining the detail of interest. The core of this transformation is related to the self-attention mechanism in the interaction between each input element to the network, relative to the other elements. Two models are proposed and tested in the study, the TransUNet and the MedicalTransformer. These, compared with some CNNs (convolutional neural networks) such as U-Net, guarantee excellent performance in the analysis and segmentation of the fire zones contained within the images of the test dataset, with F1-score equal to 97.7% and 96%. Xinguo Hou et al. [115] proposes an alternative method, based on GAN (Generative Adversarial Network), to effectively recognize and outline the fire, through a process of analysis and segmentation of the image, based on three main components of this algorithm: the generator, the discriminator and the mask. The generator is used to analyze the image and through using the mask, start the segmentation process on the flame to recognize the fire. The discriminator, on the other hand, has the task of attesting the actual presence of a flame inside the image, once the process started by the generator has ended.

Park et al. [116] addressed the challenge of the lack of wildfire occurred image datasets by employing generative adversarial networks (GAN) and weakly supervised object localization (WSOL). Their study aimed to create synthetic wildfire images with various shapes by inserting damage into free-wildfire images. These synthesized images can be utilized as training data for object detection, thereby enabling the training of deep learning models in environments where mis-detection may occur due to factors such as distance from the camera or objects resembling flame and smoke. This research contributes to the field of early detection and monitoring of wildfires using artificial intelligence and computer vision. Aslan et al. [117] proposed a vision-based method for real-time early detection of wildfire smoke using Deep Convolutional Generative Adversarial Neural Networks (DC-GANs). Their approach addresses the challenge of limited labeled data in supervised learning by employing a two-stage training framework. Additionally, a motion-based transformation of images is integrated as a pre-processing step to capture the temporal evolution of smoke. Experimental results demonstrate the effectiveness of their method in real-time smoke detection with minimal false positive rates. Jiang et al. [118] introduced a wildfire spread model implementing an Irregular Graph Network (IGN). This adaptive approach effectively encodes complex regions with dense nodes and simpler regions with sparse nodes. Comparative experiments with widely used fire simulation models were conducted on a real wildfire in Getty, California, USA. The results demonstrate that the IGN model accurately and explicitly captures the spatiotemporal characteristics of wildfire spread in a novel graph form while maintaining competitive simulation refinement and computational efficiency. Li et al. [119] designed a Recursive Bidirectional Feature Pyramid Network (RBiFPN) incorporating it into the YOLOV5 framework, to better distinguish subtle differences between clouds and smoke. They also utilize Swin Transformer to replace the classification head, enhancing the network’s capability to model local and global features by adapting the receptive fields to the size of smoke regions. Experimental results on a dataset with various interference objects demonstrate that their proposed model achieves higher performance in detecting wildfire smoke compared to state-of-the-art methods.

## 5. Open Problems and Promising Research Directions

Forest fires pose a significant threat to ecosystems, wildlife, and human lives. Over the years, various advancements have been made in the fight against forest fires. This paper explores the developments in the detection and monitoring of forest fires using WSNs and video detection techniques. Additionally, it examines the crucial role that ML and AI play in fire detection and prevention. These technologies have significantly improved the speed and accuracy of fire detection and monitoring, allowing for prompt response and effective resource allocation. As we continue to develop and refine these technologies, the potential for preventing and minimizing the impact of forest fires grows. However, challenges remain, including the need for robust and reliable communication networks, data integration, and overcoming false alarms. The existing early detection and prediction systems for wildfires face several limitations that need to be addressed. About coverage limitations, current systems may have gaps in coverage, particularly in remote or densely forested areas, where detection is challenging. Regarding false alarm rates, some systems may suffer from a high false alarm rate, leading to unnecessary deployment of resources and decreased public trust in the system. Delays in data processing and timely data processing and dissemination of warnings to relevant authorities and communities are critical. However, there can be delays in data processing, analysis, and decision-making, reducing the effectiveness of early warning systems. By leveraging the power of technology and collaboration between researchers, firefighters, and policymakers, we can continue to make significant strides in protecting our forests and communities from the devastating effects of wildfires. Future advancements may focus on developing more advanced fire detection systems that can rapidly identify the presence of fires and accurately predict their behavior. These systems could employ advanced sensors, machine learning algorithms, and real-time data analysis to enhance early detection and improve response times. Moreover, improved fire modeling techniques can help predict fire behavior more accurately. Integration of Various Data Sources and Technologies represents a further and crucial challenge to wildfire prevention. To enhance the effectiveness of early detection and prediction systems, it is essential to integrate diverse data sources and technologies. One approach is the fusion of satellite, sensor network, and weather data. By combining data from satellites, ground-based sensor networks, and weather stations, a more comprehensive and accurate understanding of fire occurrence, spread, and behavior can be achieved. This integration allows for a broader coverage area and enables a more detailed analysis of environmental conditions relevant to wildfires. Another integration aspect involves combining remote sensing techniques with ground-based monitoring systems. By incorporating aerial and satellite-based remote sensing methods with on-ground monitoring systems, such as cameras and ground sensors, the accuracy and coverage of early detection systems can be significantly enhanced. This combination enables a multi-modal approach that captures both large-scale and localized information about wildfire activities. By integrating various data sources such as weather patterns, topography, fuel moisture content, and historical fire data, predictive models can provide valuable insights into fire spread, intensity, and potential impacts. This information can assist in proactive fire management and resource allocation. The enhancing prediction accuracy and uncertainty estimation could be obtained improving the accuracy and reliability of wildfire prediction models is crucial for making effective decisions in wildfire management. There are two key strategies to achieve this goal. Firstly, it is essential to enhance model calibration and validation techniques. Properly calibrating and validating prediction models against historical wildfire data ensures their accuracy. By refining the models based on past fire behavior patterns and outcomes, their performance can be improved for future predictions. Secondly, addressing uncertainties in data inputs and model assumptions is vital. Wildfire prediction models heavily rely on environmental and weather data, which inherently have uncertainties. Accounting for these uncertainties in data inputs and model assumptions, such as through probabilistic modeling or ensemble approaches, can provide more robust predictions. Additionally, incorporating uncertainty estimation techniques into the models can help convey the confidence level associated with the predictions.

Future developments may focus on improving communication and information systems during fire incidents. This could include the implementation of robust and resilient communication networks, satellite imaging for real-time fire mapping, and the use of mobile applications to facilitate real-time data sharing among firefighters and emergency response teams. Advancements in technology offer new possibilities for improving early detection and prediction systems. One area of continuous exploration is the role of AI, IoT, and big data analytics. Leveraging AI techniques, machine learning algorithms, and big data analytics can enable more efficient processing and analysis of large volumes of data. These technologies can extract valuable insights from various data sources, enhance data fusion techniques, and improve the accuracy of predictions, ultimately leading to more timely warnings and improved decision-making. Additionally, the potential of unmanned aerial vehicles (UAVs) equipped with remote sensing technologies can be harnessed for data collection. UAVs provide real-time, high-resolution data that can aid in rapid and precise monitoring of wildfire activities. These aerial platforms can capture detailed information about fire behavior, fuel conditions, and environmental factors, contributing to more accurate early detection and prediction systems. Further developments should focus on strengthening public awareness and education regarding fire prevention and preparedness. This may involve community outreach programs, educational campaigns and the promotion of fire-safe practices. Empowering individuals with knowledge and resources can significantly contribute to reducing fire incidents and minimizing their impact. Moreover, the integration of social and behavioral factors in early detection and prediction systems can enhance their effectiveness and impact on wildfire management. One aspect is incorporating human behavior and evacuation dynamics into the models. Understanding how people respond to early warnings and evacuation orders during wildfire incidents can improve evacuation planning and decision-making. By simulating human behavior and evacuation patterns, models can provide insights into evacuation effectiveness, potential bottlenecks, and areas that require additional attention in emergency planning. Furthermore, understanding public response and decision-making processes is crucial. Factors such as risk perception, information dissemination, and community engagement play a significant role in wildfire management. Considering these factors allows for the development of more effective communication strategies, public awareness campaigns, and community-based initiatives, fostering a proactive approach to wildfire prevention and response. In conclusion, addressing the challenges in early detection and prediction of wildfires requires integrating multiple data sources, improving prediction accuracy and uncertainty estimation, considering social and behavioral factors, and exploring emerging technologies. By embracing these directions, the effectiveness of early detection and prediction systems can be enhanced, leading to improved wildfire management and reduced impact on ecosystems and communities.

## Figures and Tables

**Figure 1 sensors-23-06635-f001:**
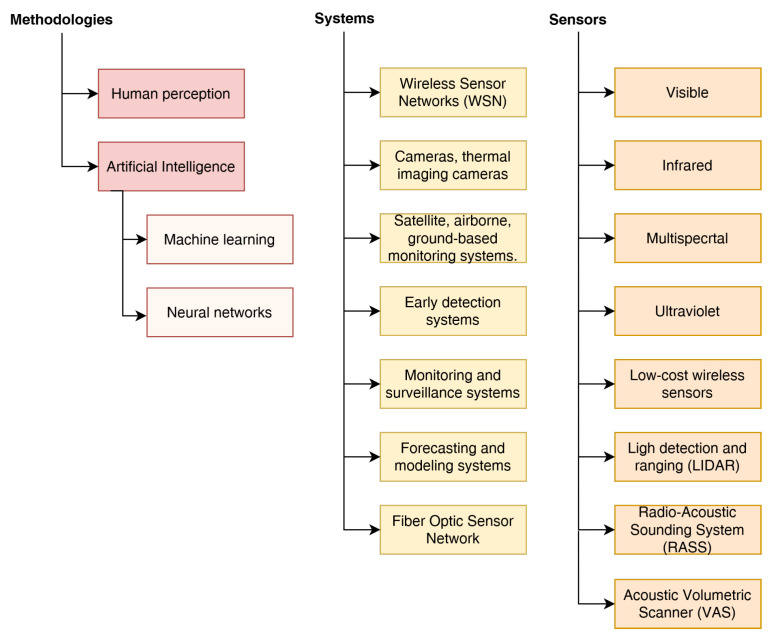
An overview of the main techniques for fire monitoring and detection.

**Figure 2 sensors-23-06635-f002:**
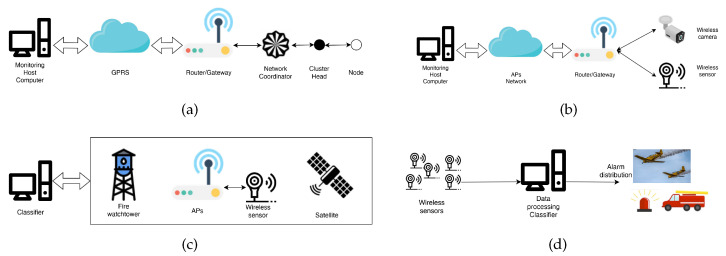
Different topologies of wireless sensor network architectures. (**a**) Topology based on dense distribution of environmental sensors; (**b**) Topology based on dense distribution of wireless sensors in addition to wireless cameras; (**c**) Classifier based on the combination of different inputs; (**d**) Data classifier based on sensor values and alert management.

**Figure 3 sensors-23-06635-f003:**
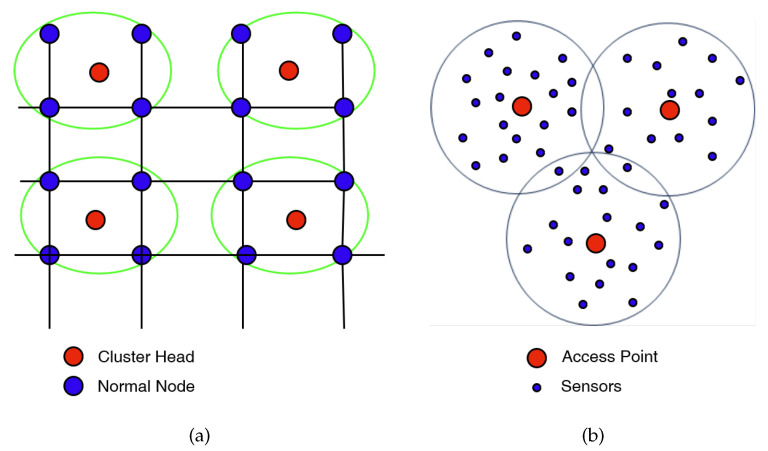
Layout for the distribution of sensor nodes. (**a**) Typical square layout with 4 sensor nodes per cluster; (**b**) Sensor distribution for coverage area.

**Figure 4 sensors-23-06635-f004:**
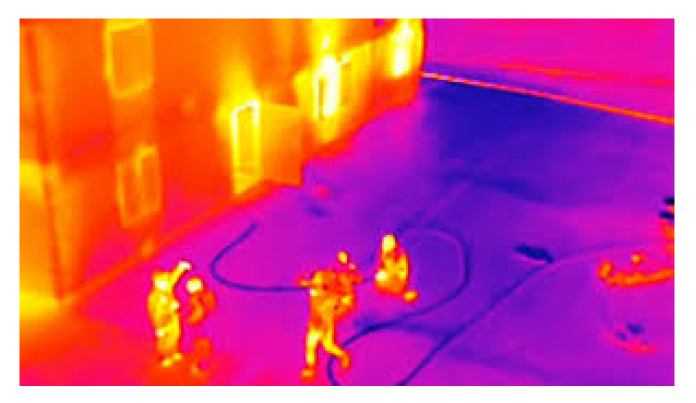
IR Camera data from UAV.

**Table 1 sensors-23-06635-t001:** Comparison table of different adopted techniques for video-based fire and smoke detection.

Paper	Color Space	Real-Time	Spectral Analysis	Moving Object	Classifier/ HMM	Wavelet	NN	Statistical Analysis	Smoke
[34]	RGB	X	X						
[37]	RGB/HSI			X		X		X	
[38]	RGB						X	X	
[39]	YUV					X			
[40]	YUV	X		X					
[41]	YUV	X				X		X	
[42]	YCbCr	X				X		X	
[43]	YCbCr/RGB		X						
[44]	Lab	X		X		X		X	
[45]	Lab							X	
[46]	Lab/YCbCr	X		X				X	
[47]	HSV			X				X	
[48]	HSV	X					X	X	
[49]	HSV/HSL			X					
[50]	HSV	X		X	HMM				
[51]	RGB/HSV	X		X				X	
[52]	RGB/HSV		X					X	
[53]	Luv		X	X				X	X
[54]	HSI	X	X					X	
[55]	FCS	X						X	
[56]		X				X			
[57]	RGB	X		X		X			
[59]		X		X	weak classifier, HMM	X			
[60]	greyscale			X		X	X	X	X
[62]		X		X	HMM			X	
[64]				X	SVM	X			
[65]			X		SVM			X	
[66]	RGB				SVM			X	
[67]				X	RAB	X			X
[69]								X	
[70]	RGB		X			X		X	
[72]	RGB			X		X	X	X	
[73,74]		X		X			X	X	
[75]							X		
[77]	RGB	X	X	X				X	X
[78]				X				X	X

**Table 2 sensors-23-06635-t002:** Resume about the various image sensors used in satellites.

Image Sensor	AVHRR	MODIS	VIIRS
Spatial Resolution	1 km	1 km	750 m
Spectral Band	6 (bands at 1 km)	36 (2 bands at 250 m, 5 bands at 500 m and 29 at 1 km resolution)	22 (16 moderate-resolution bands at 750 m, 5 image resolution at 750 m and 1 pancromatic day/night at 750 m)
Advantages	infrared	more various channels for more perfect earth mapping	The pancromatic day/night band gives an opportunity to take more accurate forest fire data
Disadvantages	Few channels for multiple image analyze in terms of resolution. To complete earth mapping it takes 102 min	To complete earth mapping it takes 99 min	To complete earth mapping it takes 50 min

**Table 3 sensors-23-06635-t003:** Comparison table between environmental data trained ML models with vs computer vision techniques.

Technique	Real-Time Detection	Reliability in Identification	Wildfire Prediction	Simulation of Fire Spread	Robustness	Complexity	Area Covered
ML models w/ environment data	X	Low	X	X	Low	Low/Medium	Medium/Large
Computer vision	X	High			High	High	Very variable

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
