# Peer review of "Advancements in Forest Fire Prevention: A Comprehensive Survey"

_sensors, 2023, doi:10.3390/s23146635_

Round 1
Reviewer 1 Report
The structure of this review is commendable as the authors have effectively presented numerous methods in a well-organized manner
The English language utilized in this paper is exceptionally proficient, requiring only minor editing to further enhance its quality.
Author Response
Dear Editor and Reviewers,
We would like to thank you all for your valuable comments and remarks, which have been very helpful to improve the quality of the submitted manuscript. We have taken every effort to address all of the issues raised and update our paper according to the offered comments. In the revised version of the manuscript the changes are highlighted with text in blue color. For more detail, please, refer to the responses below. We hope that you will find the revision satisfactory.
Sincerely yours,
Francesco Carta, Chiara Zidda, Martina Putzu, Daniele Loru, Matteo Anedda, and Daniele Giusto

Reviewer 2 Report
The paper intend to be a comprehensive survey of detection, monitoring, and prevention of forest fires techniques.
After short introduction, techniques based on wireless sensory networks and video based detection techniques are described. Particular attention is given to the role of machine learning and artificial intelligence in the detection and prevention of fires.
To be a comprehensive survey the paper need major improvements concerning topics listed bellow:
- Although authors mention detection, monitoring, and prevention in paper almost nothing about prevention is given. All described techniques deals with detection and monitoring. Wildfire prevention is completely another topic. I sugest authors to analyse papers from previous International Conference on Forest Fire Research, Coimbra, Portugal (Open access, every 4 years, the last one in 2022 was IX conference). On every conference there were sections dedicated to wildfire detection, as well as wildfire prevention. Also I suggest authors to analyse certain texts concerning these topics in Encyclopedia of Wildfires and Wildland-Urban Interface (WUI) Fires (ed. Samuel L. Manzello).
- Introductory figure (Figure 1) is taken from one of references, but it covers only video based detection techniques and authors also discuss techniques based on wireless sensory networks. Therefore I suggest that as an introductory figure authors have to create, draw their own image taking into account all techniques proposed and used up to date.
- Concerning all proposed techniques authors have missed a lot of them (Optical spectrometry, Light detection and ranging (LIDAR), Radio-Acoustic Sounding System (RASS) for remote temperature measurements and thermal sensing of a particular forest region, Acoustic Volumetric Scanner (VAS), Fiber Optic Sensor Network (FOSN) to exotic proposals where animals have to be used as mobile biological sensors equipped with sensor devices.) There ware some papers dealing with these topics in Proceeding of International Conference on Forest Fire Research, Coimbra, Portugal.
- Nothing about theoretical background of wildfire detection and monitoring was mentioned and good theoretical description of such systems is based on the observer theory where observers and observer networks were formally mathematically described.
- In ground video based wildfire detection techniques before AI and ML techniques a lot of advanced features has been proposed. There were some survey since 2014 not analysed by authors of manuscript.
- In one part of their work authors discuss the use of different color spaces in wildfire detection but not all color spaces were analysed, and there ware a lot of resarch in this field.
- Authors also discuss use of AI and ML in wildfire detection, but not analyse all survey papers in this field. AI and ML techniques are particularly in use for wildfire detection in the last 15 years.
- To be a comprehensive survey all important papers has to be analysed, and fome one of seminal papers from 1989: Cappellini, V., Mattii, L., Mecocci, A. "An intelligent system for automatic fire detection in forests." Third Int. Conference on Image Processing and its Applications. 1989. 563-570. to more recent one. In Reference list there is only one paper from 2022 and no one paper from 2023, and as I know in the last two years wildfire detection, particularly using AI and ML techniques is quite a popular topic.
Last but not least a term 'wildfires' is maybe more appropriate instead of 'forest fires'.
Author Response

(The authors gave the same response as above.)

Reviewer 3 Report
This paper investigates advancements in detecting and monitoring forest fires using wireless sensor networks and video detection technology. Furthermore, it evaluates the essential role of machine learning and artificial intelligence in fire detection and prevention. The authors assess the merits and demerits of these technologies and provide insights into the future.
The insufficiencies and areas in need of improvement in this article can be summarized as follows:
1. The title may contain a typographical error. Changing "Fire Forest Prevention" to "Forest Fire Prevention" would make the title more accurate and clear.
2. The introduction section of the article requires modifications regarding background information and structure. For instance, while various factors contribute to forest and land area losses, as an article focusing on forest fire prevention, it should underscore the significant threat that forest fires pose to forests and land. Moreover, the introduction should arrange the text in the sequence of the different forest fire prevention technologies discussed later in the article. Additionally, the article should explicitly elucidate the relationship between the three technologies and their distinct roles in fire detection and prevention.
3. Figure 1 of the article classifies fire prevention systems into three categories: Terrestrial-based, Aerial-based, and Satellite-based. However, the article's description of the corresponding technology deviates from these perspectives. It is highly recommended that the article includes more citations and syntheses following these three viewpoints to enhance the article's rationale and comprehensiveness. Moreover, the primary focus of this paper is on forest fire prevention techniques that are predominantly associated with machine learning technologies. Consequently, it is advisable for the author to integrate research findings on alternative approaches, such as those presented in 10.3390/f14010102.
4. The title of Section 4 in this paper should be revised to "The Role of Traditional Machine Learning and Deep Learning in Fire Detection and Prevention." This change better corresponds with the section's content.
5. The references in the section on traditional machine learning might be inadequate to support the discussion. For instance, this section concentrates on machine learning applications in fire detection and warning, but regarding fire prevention, there are numerous applications such as wildfire risk prediction (or wildfire susceptibility prediction), fire behavior modeling (e.g., fire spread prediction), and emergency response to fires (e.g., resource dispatching). It is advised to expand this section based on these aspects. Here are a few examples:
10.3390/f13111826
10.3390/rs14174362
10.3390/f13081332
6. Likewise, the adequacy and scope of section 4.2 need enhancement. Many advanced deep networks and models have shown promising results in forest fire prevention. It is recommended to follow up and cite research findings on advanced deep networks and models in this area, such as generative adversarial networks(GANs), graph neural networks(GNNs), and Transformers.
In the entire article, some grammar errors or unclear expressions need to be checked and corrected. For example, some sentences may need to be split, and some sentences have redundant or repetitive phrases that need to be corrected to improve readability.
Author Response

(The authors gave the same response as above.)

Reviewer 4 Report
This is a well-written paper, in a very important subject. However, I understand that it can be improved in several directions before it can be considered for publication again.
- As this work deals with a very practical problem, I miss a more detailed discussion on current forest fire prevention techniques and system. By that I mean systems currently in use around the world. A detailed discussion on the methods that are being used and their limitations would be very welcome.
- Then, the academic part could be improved as well. The literature review can be improved. Please consider including related works from high quality conferences and journal. Moreover, the literature review can be updated.
- Finally, a very detailed discussion on the open problems and promising research directions is very important in a survey paper.
Although the quality of the English language is fine, a professional proof-reading would be welcome.
Author Response

(The authors gave the same response as above.)

Round 2
Reviewer 2 Report
The authors significantly improved their work.
Reviewer 3 Report
The authors undertook detailed revisions and additions in response to the key points raised in the comments, providing further information to support the article. From the revised manuscript, it can be seen that the authors have made detailed additions to address the issues raised in the comments, greatly enhancing the background and comprehensiveness of the article. As a review-type paper, such additions are necessary, and the content added by the authors is comprehensive and substantial.
It is clear from these revisions that the authors have taken the feedback into serious consideration. They have made a concerted effort to address all concerns and suggestions, resulting in a more robust and well-supported manuscript.
Reviewer 4 Report
Thanks for the detailed answers. I have no more comments.